# Kinesin-14 motors drive a right-handed helical motion of antiparallel microtubules around each other

Aniruddha Mitra[1,2,4], Laura Meißner [1], Rojapriyadharshini Gandhimathi[1,5], Roman Renger[1,2], Felix Ruhnow[1,6] & Stefan Diez [1,2,3✉]

Within the mitotic spindle, kinesin motors cross-link and slide overlapping microtubules. Some of these motors exhibit off-axis power strokes, but their impact on motility and force generation in microtubule overlaps has not been investigated. Here, we develop and utilize a three-dimensional in vitro motility assay to explore kinesin-14, Ncd, driven sliding of cross-linked microtubules. We observe that free microtubules, sliding on suspended microtubules, not only rotate around their own axis but also move around the suspended microtubules with right-handed helical trajectories. Importantly, the associated torque is large enough to cause microtubule twisting and coiling. Further, our technique allows us to measure the in situ spatial extension of the motors between cross-linked microtubules to be about 20 nm. We argue that the capability of microtubule-crosslinking kinesins to cause helical motion of overlapping microtubules around each other allows for flexible filament organization, road-block circumvention and torque generation in the mitotic spindle.

[1] B CUBE—Center for Molecular Bioengineering, Technische Universität Dresden, 01307 Dresden, Germany. [2] Max Planck Institute of Molecular Cell Biology and Genetics, 01307 Dresden, Germany. [3] Cluster of Excellence Physics of Life, Technische Universität Dresden, 01062 Dresden, Germany. [4] Present address: Department of Physics and Astronomy, and LaserLaB Amsterdam, Vrije Universiteit Amsterdam, 1081HV Amsterdam, The Netherlands. [5] Present address: Institut für Biochemie, ETH Zürich, 8093 Zürich, Switzerland. [6] Present address: School of Biosciences, University of Melbourne, Parkville, VIC 3010, Australia. ✉email: stefan.diez@tu-dresden.de

The mitotic spindle is a complex subcellular machinery that segregates chromosomes during eukaryotic cell division. Several in vivo and in vitro studies provide us with a consolidated two-dimensional (2D) model, detailing the mechanisms employed by dynamic microtubules, kinesins and microtubule-associated proteins (MAPs) to assemble, maintain and disassemble the spindle in order to coordinate chromosome segregation[1,2]. However, the spindle is a three-dimensional (3D) structure and interesting mechanical details may emerge in the third dimension, which are not just intuitive extensions of the 2D model. For instance, previous studies have shown that the mitotic spindle is twisted with a distinct left-handed chirality[3–6].

Interestingly several mitotic kinesins, like kinesin-5[7], kinesin-8[8,9] and kinesin-14[10,11], have been shown to exhibit off-axis components in their stepping behavior, i.e., they do not move strictly parallel to the longitudinal microtubule axis. These mitotic kinesins are microtubule cross-linkers involved in connecting and sliding microtubules[12–14]. To address if their off-axis (i.e., axial) motion translates to cross-linked microtubules, we explore the 3D motion of microtubules propelled along each other by the kinesin-14. Kinesin-14 is a non-processive, minus-end directed motor with an additional diffusive microtubule-interaction site in its N-terminal tail domain that enables the motor to cross-link and slide anti-parallel microtubules[12]. Specifically, we suspend long microtubules on nano-fabricated ridges[9,15] and track the motion of cross-linked, shorter microtubules along the long ones. Besides being able to measure the in situ spatial extension of the motors between the cross-linked microtubules, we discover that antiparallel microtubules both rotate around their own axis and helically move around each other in a right-handed manner. Importantly, the torque resulting from the underlying off-axis motor forces is large enough to cause microtubule twisting and coiling.

## Results

**Kinesin-14 motors induce helical sliding of microtubules**. To study the uninhibited sliding motion of cross-linked microtubules driven by *Drosophila melanogaster* kinesin-14 motors, Ncd, we suspended parts of long "template" microtubules such that short cross-linked "transport" microtubules could access the entire lattice of the suspended template microtubules. As illustrated in Fig. 1a, rhodamine-labeled template microtubules were tautly fixed on optically transparent polymer ridges (370 nm high and 2 or 5 μm wide) separated by 10 μm wide valleys via anti-rhodamine antibodies, as developed in previous studies[9,15]. Atto647n-labeled transport microtubules were cross-linked to the template microtubules via full length GFP-Ncd (4 nM if not otherwise noted), hereafter referred to as Ncd, in ADP. After the addition of ATP, antiparallel transport microtubules started to slide along the template microtubules while parallel transport microtubules remained locked in their position. To observe the position of the transport microtubules with respect to the template microtubules, the microtubule positions were tracked in 2D individually in the rhodamine and Atto647n fluorescence channels using FIESTA[16] and the distance of the center points of the transport microtubules from the tracked center line of the template microtubules, referred to as the sideways distance, were obtained (Fig. 1b). Since the imaging set-up does not provide information about the third dimension, the sideways distances provide the *xy*-projected distances between the cross-linked microtubules, which are maximum when the transport microtubules are at the extreme left (or right) of the template microtubules and zero when the transport microtubules are on top (or bottom) of the template microtubules.

To interpret the 3D motion of antiparallel transport microtubules, the sideways distance was plotted with respect to the distance traveled in the longitudinal direction of the template microtubules. Figure 1c (and Supplementary Movie 1) shows an exemplary sliding event, where a transport microtubule moved along a template microtubule suspended over two valleys with a 2 μm wide ridge in between. In the valley regions, the transport microtubule followed a helical trajectory with a pitch of $1.3 \pm 0.1\,\mu m$ (mean ± SD; $N = 7$ complete rotations). When the transport microtubule encountered the ridge, its helical motion was blocked without significant impediment to the longitudinal motion, and it stayed at the right-hand side of the template microtubule. The transport microtubule only resumed its helical motion after the entire microtubule (1.8 μm long) had left the ridge. This indicates that surface immobilization of the template microtubule on the ridge hindered the helical motion of the transport microtubule. Since the helical motion of the transport microtubule on the ridge was locked on the right-hand side of the template microtubule, we infer that the helical motion is right-handed (or clockwise in the direction of motion).

We analyzed 125 microtubule-microtubule sliding events on suspended template microtubules. Antiparallel transport microtubules (94 events) exhibited robust helical trajectories in the valley regions along suspended template microtubules (see 30 example events in Fig. 2a and all 94 rotating events in Supplementary Fig. 1a). Interestingly, while the helical pitches within individual events were similar, they sizably varied between events (ranging from 0.5 to 3 μm as seen in Supplementary Fig. 1b; median helical pitch $1.6 \pm 0.2\,\mu m$; see methods for analysis of distributions with error estimation) often even on the same template microtubule. There was no evident correlation between the helical pitches and the sliding velocities (Supplementary Fig. 1c). Also, the helical pitches were not correlated with the lengths of the transport microtubules, though shorter microtubules exhibited larger variations in helical pitches (Supplementary Fig. 1d). When sliding on the ridge-immobilized parts of the template microtubules, transport microtubules were almost always on the right-hand side of the template microtubules (Fig. 2b), with a mean sideways distance of $-41.6 \pm 0.4\,nm$ ($N = 63$). This clearly indicates that the helical motion induced by Ncd is right-handed. Only in a couple of events the transport microtubules moved around the surface-immobilized template microtubules, presumably at locations of statistically low densities of anti-rhodamine antibodies. These observations are consistent with 2D sliding motility assays performed on unstructured glass coverslips (Supplementary Fig. 2a, example event in Supplementary Movie 2). In this assay geometry, transport microtubules moved either along the right-hand side of the template microtubules or flipped around the template microtubules with aperiodic helical motion with no clear dependence on motor density (Supplementary Fig. 2b). The mean sideways distance was $-44.6 \pm 0.1\,nm$ ($N = 98$) and upon removing the flip events became $-45.5 \pm 0.1\,nm$ ($N = 83$). At low Ncd concentration (0.02 nM), the sideways motion of transport microtubules became erratic. However, there was still a bias to the right-hand side with a mean sideways distance of $-26.8 \pm 0.7\,nm$ ($N = 24$). Parallel transport microtubules (31 events) were found to be locked, both in their longitudinal and axial direction, and did not exhibit any preference for which side of the template microtubule they were bound to (mean sideways distance $13.7 \pm 1.0\,nm$; $N = 31$, Fig. 2c). Rather, the distribution of sideways distances was bimodal, consistent with the distribution of microtubule protofilaments projected in 2D and an equal probability of transport microtubules interacting randomly with any protofilament of the template microtubules, providing peaks at about ±45 nm. In summary, transport microtubules that were

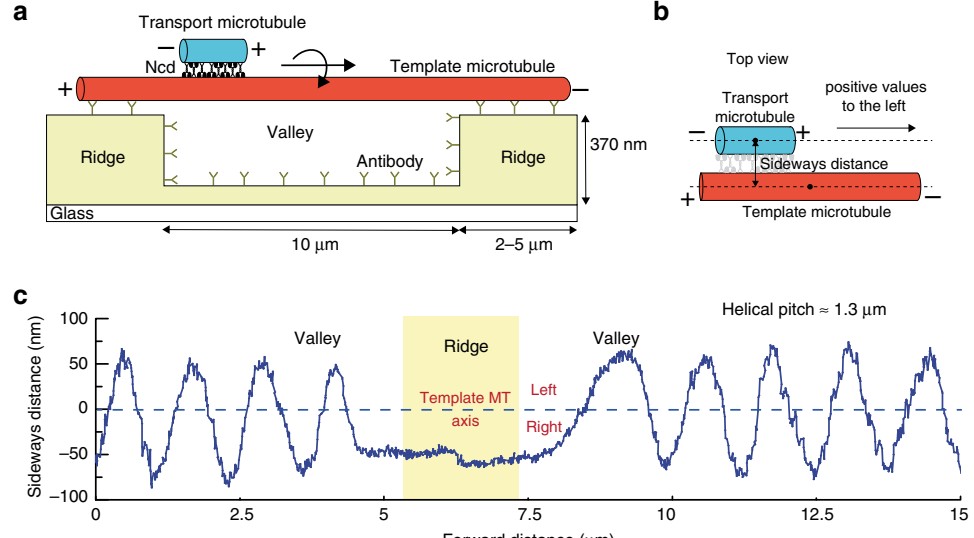

**Fig. 1 3D sliding of transport microtubules driven by Ncd on suspended template microtubules. a** Schematic representation of a suspended, rhodamine-labeled, template microtubule immobilized on optically transparent polymer ridges by anti-rhodamine antibodies. Atto647n-labeled transport microtubules are capable of freely accessing the 3D lattice of the template microtubule between the ridges (in the region referred to as valley), as they slide along them driven by GFP-labeled Ncd motors. **b** The perpendicular distance of the center point of the transport microtubule from the tracked center line of the template microtubule provides the sideways distance. Positive values are arbitrarily assigned to positions on the left. **c** Sideways distance for an example antiparallel transport microtubule as it slides along a template microtubule over two valley regions (see also Supplementary Movie 1). In both valley regions, the 1.8 μm long transport microtubule exhibited a helical motion around the suspended template microtubule, with a helical pitch of 1.3 ± 0.1 μm ($N = 7$ rotations, mean ± SD). At the ridge, where the template microtubule is surface-immobilized, the transport microtubule remained on the right-hand side of the template microtubule, indicating that the direction of helical motion is right-handed.

antiparallel to the template microtubules showed robust right-handed helical motion on suspended template microtubules with helical pitches of 0.5–3 μm and their helical motion was hindered on surface-immobilized template microtubules. In contrast, parallel transport microtubules were locked in the longitudinal as well as the axial direction.

**Extension of kinesin-14 in microtubule overlaps is ~20 nm.** Intriguingly, our technique additionally allows us to gauge the in situ spatial extension of the motors between sliding microtubules with nanometer precision. Toward this end, we determined the diameter of the helical trajectories (distance between adjacent maxima and minima in the sideways distance) for all 3D sliding events with at least two complete helical turns. We obtained a mean value of 86.2 ± 4.4 nm ($N = 94$, Fig. 3a) yielding an extension of the Ncd motors of about 18 nm (diameter of template microtubule [25 nm] + 2 × radius of transport microtubule [25 nm] + 2 × in situ extension of Ncd motors [2 × 18] = diameter of the helical path [86 nm]; Fig. 3c). The diameter of the helical path showed no correlation with neither helical pitch nor sliding velocity (Supplementary Fig. 3). Another estimate for the in situ extension of Ncd motors was obtained from the sideways distance of sliding events that remained on the extreme right-hand side of surface-immobilized template microtubules (45.5 ± 0.1 nm; $N = 83$; Fig. 3b). This value yields an extension of the Ncd motors of about 21 nm (radius of template microtubule [12.5 nm] + radius of transport microtubule [12.5 nm] + in situ extension of Ncd motors [21] = extreme right-hand sideways distance [46 nm]; Fig. 3c). Therefore, we estimated the in situ extension of Ncd motors between cross-linked microtubules to be about 18–21 nm.

**Motor density determines rotational pitch of microtubules.** To elucidate the origin of the variation in the helical pitches of

transport microtubules on suspended template microtubules (see Fig. 2a, Supplementary Fig. 1a, b) beyond the sliding velocity and the microtubule length (Supplementary Fig. 1c, d) we tested for the influence of the Ncd motor density. Unfortunately, direct measurements in our 3D sliding motility assays on suspended template microtubules were hampered for the following reasons: (i) It was not possible to reliably quantify the motor density from the intensity of the GFP-signal of the motors due to the auto-fluorescence of the microfabricated polymer structures. (ii) Lowering the Ncd concentration (to 0.4 nM) caused most trans-port microtubules to engage in an erratic helical motion (with only few events exhibiting complete rotations) even though the longitudinal motion along the template microtubules axis was not impeded. (iii) Increasing the Ncd concentration (to 40 nM) led to unspecific motor binding to the polymer structures causing the template microtubules to glide and detach. We therefore turned to microtubule rotation measurements (i.e., the investigation of the rotational motion of microtubules around their own axis) in 2D motility assays on unstructured surfaces. First, we performed gliding motility assays on reflective silicon wafers, where rhodamine-speckled microtubules driven by Ncd motors (attached to the surface via Fab fragments and Anti-GFP anti-bodies) were imaged using fluorescence interference contrast (FLIC) microscopy[17]. The recorded intensities of the speckles fluctuated periodically due to changes in speckle height with respect to the reflective surface, indicative of microtubule rota-tions (Fig. 4a). We observed that the rotational pitch and the sliding velocity of the microtubules reduced upon increasing the surface density of Ncd motors (Fig. 4b, Supplementary Fig. 4a). Secondly, we performed FLIC-based sliding motility assays on silicon wafers with rhodamine-speckled transport microtubules (Fig. 4c). While the helical motion of sliding transport micro-tubules around the surface-immobilized template microtubules was mostly blocked (as expected, see also Supplementary Fig. 2), we observed that the transport microtubules still rotated robustly

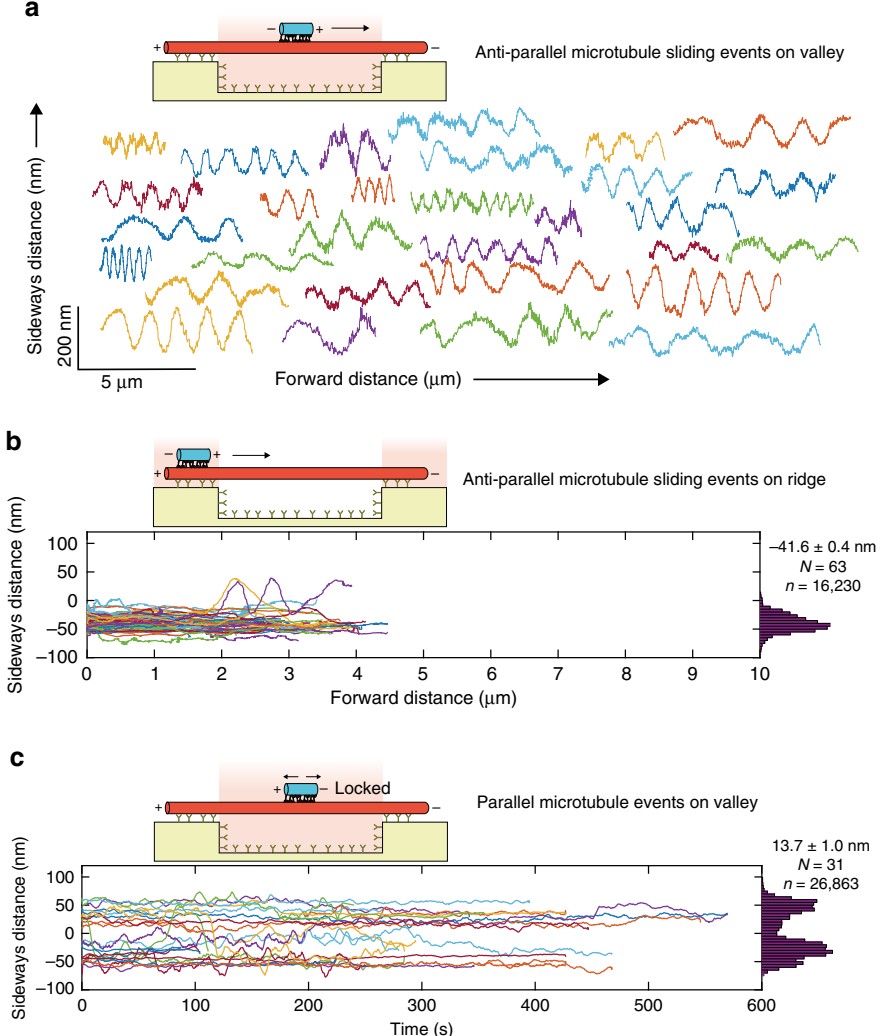

**Fig. 2 Trajectories of transport microtubules antiparallel and parallel to the template microtubules. a** Sideways distance of 30 example antiparallel transport microtubule (out of 94 events; all events shown in Supplementary Fig. 1A) driven by Ncd along the suspended parts of template microtubules. Each transport microtubule exhibited a robust helical motion with relatively constant helical pitch. Between various transport microtubules, helical pitches varied significantly between 0.5 and 3 μm. **b** Sideways distance of 63 antiparallel transport microtubules along the ridge-immobilized parts of the template microtubules. Most transport microtubules remained on the right-hand side of the template microtubule while a couple of microtubules managed to squeeze under the template microtubules to perform complete rotations. The average sideways distance was −41.6 ± 0.4 nm ($N = 63$ events; $n = 16,230$ data points; see methods for analysis of distributions with error estimation). **c** Sideways distance (plotted with respect to time) of parallel transport microtubules along the suspended parts of template microtubules. The average sideways distance was 13.7 ± 1.0 nm ($N = 31$ events; $n = 26,863$ data points). All events showed very little (<300 nm) or no motion in the longitudinal direction. Transport microtubules were additionally locked in the axial direction with no preference for either side of the template microtubule. Only events on template microtubules having one or more antiparallel microtubule sliding events were chosen, in order to define whether the transport microtubule is on the left or right-hand side of the template microtubule. In **b** and **c**, the smoothened trajectories are plotted (rolling frame averaged over 20 frames).

around their own axis (Fig. 4c, Supplementary Movie 3). At a low Ncd concentration (0.6 nM) only few transport microtubules rotated periodically (only 7 out of 35 events analyzed), while at higher Ncd concentrations (2.4 and 6 nM Ncd) almost all transport microtubules rotated robustly. This behavior was confirmed when the Ncd concentration (initially 0.6 nM) was increased to 6 nM in situ during imaging. Transport microtubules which had not rotated periodically at 0.6 nM Ncd started to rotate robustly with short rotational pitches after switching to 6 nM Ncd (Supplementary Fig. 4c). Further, we observed that the rotational pitch and sliding velocity of the transport microtubules reduced upon increasing the Ncd concentration (Fig. 4d, Supplementary Fig. 4b). In summary, the rotational pitch of Ncd-driven gliding and sliding microtubules is influenced by motor density.

Interestingly, the rotational pitches of microtubules in these 2D gliding and sliding motility assays (0.5–2 μm) were in the same range as the helical pitches of microtubules in 3D sliding motility assays using suspended template microtubules.

**Kinesin-14 motors generate torque in microtubule overlaps.** Having shown that transport microtubules rotate around their own axis and move helically around the suspended template microtubules, we asked if the underlying off-axis motor forces would generate a torque that was high enough to cause twisting deformations of microtubules. Towards this end, we performed 2D sliding motility assays (Ncd concentration of 4 nM) on unstructured glass surfaces with long transport microtubules and short, surface-immobilized template microtubules. We focused on

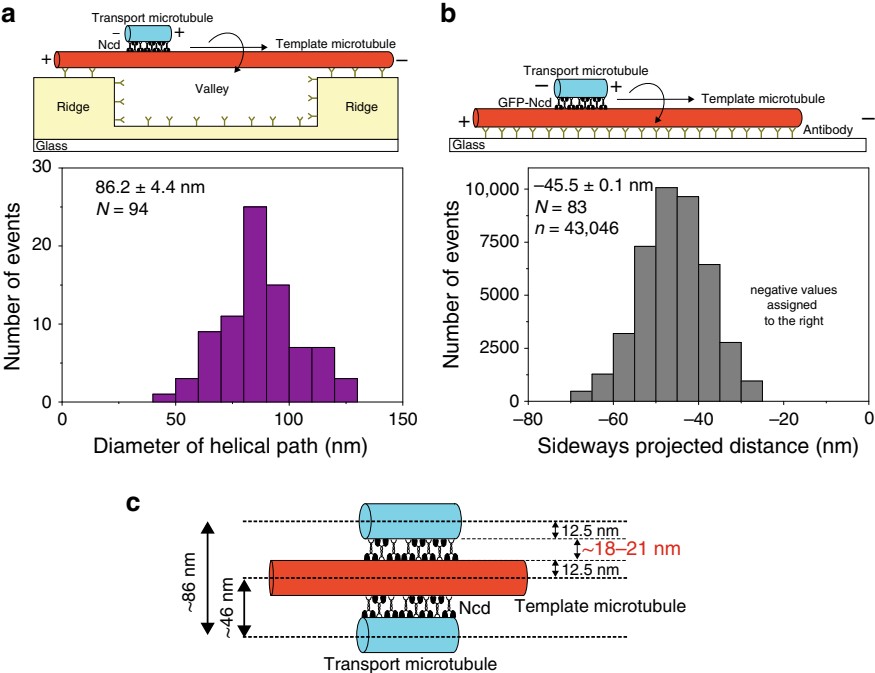

**Fig. 3 Spatial extension of Ncd motors between sliding microtubules. a** Histogram corresponding to the diameter of the helical trajectories taken by antiparallel transport microtubules rotating around suspended template microtubules (3D sliding motility assay). The average diameter was 86.2 ± 4.4 nm ($N = 94$ events; see methods for analysis of distributions with error estimation). The mean diameter of the helical paths was calculated from the peak-to-peak sideways distances obtained from the trajectories of antiparallel transport microtubules. **b** Histogram corresponding to the sideways distance of antiparallel transport microtubules sliding along the right-hand side of surface-immobilized template microtubule (2D sliding motility assay). The average sideways distance was −45.5 ± 0.1 nm ($N = 83$ events; $n = 43,046$ data points). **c** As shown in the illustration, considering the diameter of the template and transport microtubule (25 nm), the in situ extension of Ncd motors was calculated to be about 18 and 21 nm for the 3D and 2D sliding motility assays, respectively.

events, where the leading end of a transport microtubule was locked in a parallel configuration on one template microtubule and the trailing end of the same transport microtubule was sliding in an antiparallel configuration on another template microtubule (Fig. 5a). We found that the central part of those transport microtubules was often twisted into loops before being coiled up into double helices with up to eight turns (see Fig. 5, Supplementary Movie 4 for an exemplary event and Supplementary Fig. 5 for time-lapse micrographs of three additional events). This indicates that the right-handed torque generated by Ncd motors in the sliding geometry (where multiple Ncd motors generate only sub-pN longitudinal forces[18]) is high enough to twist and coil microtubules.

## Discussion

The longitudinal motion of motors and cargo along surface-immobilized microtubules has been studied extensively in 2D in vitro motility assays. However, recent studies revealed that, in order to explore the complete 3D motion, the microtubules need to be suspended[9,15,19,20]. This also became evident in our experiments where individual transport microtubules could helically move around the suspended parts of template microtubules, but were held on the right-hand side of the surface-immobilized parts of the template microtubules (Fig. 2b, Supplementary Fig. 2b). Because microtubules were always held on the right-hand side of the template microtubules, we reason that the helical motion on suspended microtubules is right-handed. The median helical pitch of 1.6 ± 0.2 μm, consistent with the values of rotational pitches of microtubules gliding on surface-bound Ncd motors[11], cannot be related to the supertwist of the microtubules (all grown in the presence of GMPCPP and

stabilized by taxol). The vast majority (about 96%[21]) of these microtubules are expected to comprise 14 protofilaments, which provide a left-handed supertwist of about 8 μm[22,23]. Conceivably, the origin of the right-handed helical motion is related to the off-axis component in the power stroke of Ncd (longitudinal stalk displacement of 9 nm accompanied by a net angular displacement of 1.1°, see ref. [11]), elucidated in previous studies exploring the rotational motion of microtubules gliding on surface-bound Ncd motors[10,11]. However, the geometry in which the motors perform their power strokes is crucially different. In previous studies, motors were bound rigidly to a planar surface (conventional surface gliding assays). In our current study, microtubules are linked to each other by motors diffusively coupled to one microtubule via their tail domains and interacting with the other microtubule via their motor domains (microtubule–microtubule sliding geometry). Strikingly, in this geometry (where motor force in the forward direction goes down significantly[18]), transport microtubules still rotate around their own longitudinal axis but additionally also move helically around the template microtubules. For parallel transport microtubules, the motors facing opposite directions between the cross-linked microtubules antagonize each other, locking the microtubules longitudinally as well as axially (Fig. 2c).

Based on our observations in FLIC measurements (Fig. 4b), we reason that transport microtubules exhibit right-handed rotations around their own axis in addition to their right-handed helical motion around suspended template microtubules (Figs. 1 and 2, see also illustration in Fig. 6a). To explain such complex movement, we carefully explored the geometry of the microtubules in the different motility assays (see Fig. 6b, c for illustrations of the transverse sections of the microtubules). In gliding motility assays (Fig. 6b), the asymmetric power stroke of Ncd results in a

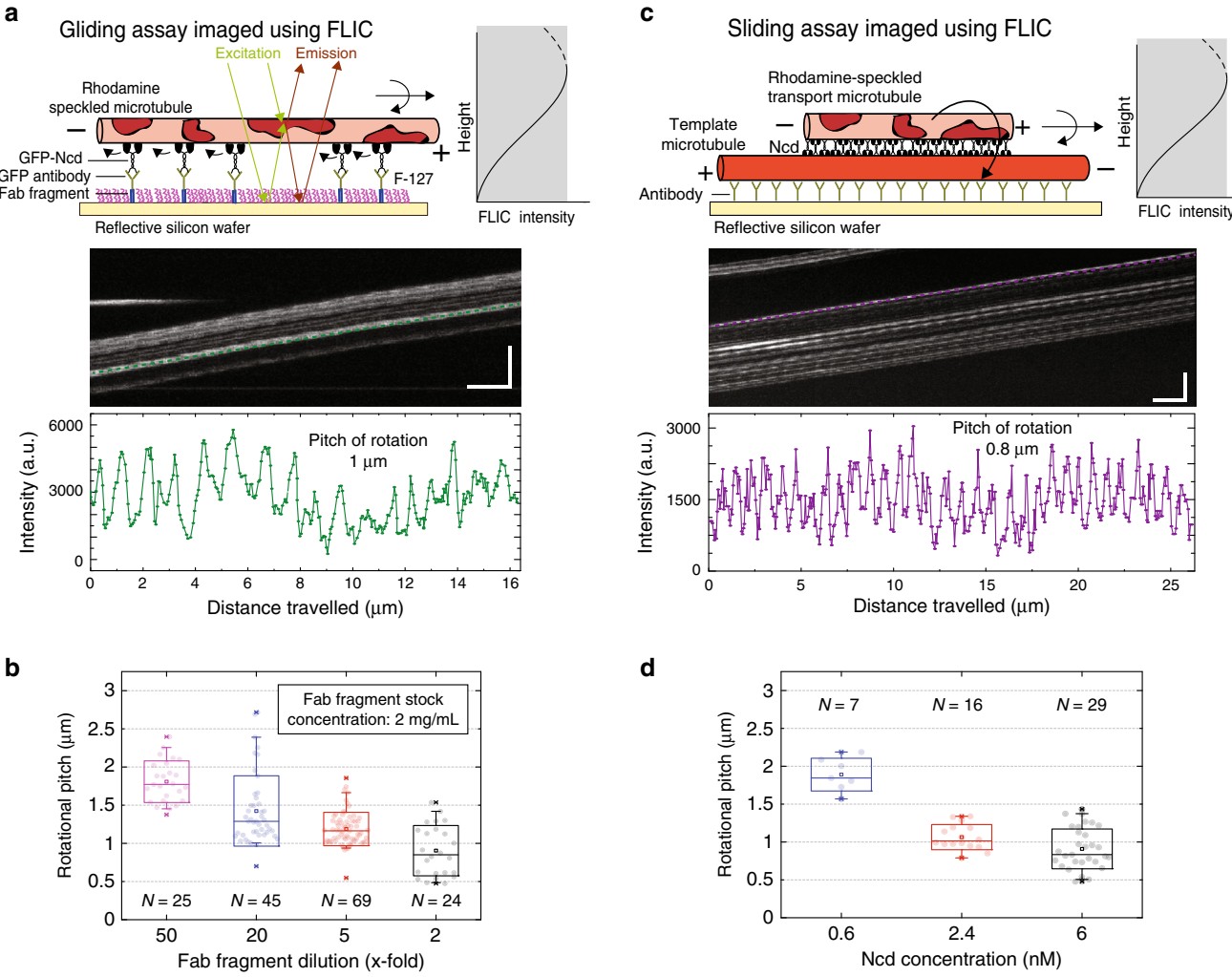

**Fig. 4 Dependence of rotational pitch of microtubules on Ncd motor density. a** Measurement of the rotational pitch of speckled microtubules gliding on a reflective silicon substrate coated with GFP-Ncd via anti-GFP antibodies bound to Fab fragments. Due to fluorescence interference contrast (FLIC), the recorded intensities of the asymmetric speckles change as a function of height above the substrate. Rotational information of gliding speckled microtubules is interpreted from the periodic variation in the recorded intensity of the speckles on the microtubule (vertical scale bar: 10 μm; horizontal scale bar: 40 s in the example kymograph, FLIC intensity profile over time for one of the speckles indicated by the green line in the kymograph). The rotational pitch of this gliding microtubule is about 1 μm. **b** The rotational pitch of the gliding microtubules was dependent on the surface density of motors. The motor density was set by changing the Fab fragment dilution (Fab fragment stock concentration of 2 mg/mL; diluted 50×, 20×, 5×, and 2×; N = 25, 45, 69, and 24 events, respectively) and ranged between 0.5 and 2.0 μm. **c** FLIC-based measurement of the rotational pitch of speckled transport microtubules sliding on surface-immobilized template microtubules (vertical scale bar: 10 μm; horizontal scale bar: 40 s in the example kymograph, FLIC intensity profile over time for one of the speckles indicated by the purple line in the kymograph). The rotational pitch of this sliding transport microtubule is about 0.8 μm (see also Supplementary Movie 3). **d** The rotational pitch of the sliding transport microtubules was dependent on the motor concentration (0.6, 2.4, and 6 nM; N = 7, 16, and 29 events, respectively) and ranged between 0.5 and 2 μm. All the rotational pitch distributions in (**b**) and (**d**) are significantly different. Supplementary Fig. 4C provides the corresponding p values obtained from two-sample two-sided Mann–Whitney U-tests. For the boxplot description, see Methods.

right-handed rotational motion of the gliding microtubule around its own axis, as also observed in previous studies[10,11]. In sliding motility assays (Fig. 6c), Ncd motor domains facing the transport microtubule are geometrically similar to the Ncd motors in the gliding geometry, expected to rotate the transport microtubule around its own axis in a right-handed manner. In contrast, Ncd motors with motor domains facing the template microtubule cannot rotate the template microtubule because it is fixed on the surface. Consequently, the off-axis power strokes of these motors are transposed to the transport microtubule, moving it helically around the template microtubule in a right-handed manner. Therefore, the two populations of Ncd motors enable the transport microtubules to undergo a combination of rotational

motion around their own axis and helical motion around the template microtubules, both motions being right-handed with pitches in a similar range (0.5–2 μm).

The sizeable variation in the observed helical pitches and velocities of transport microtubules in the 3D sliding assays cannot be entirely attributed to motor stochasticity as microtubule sliding involves multiple motors. In addition, we did not observe any correlation of the helical pitches and velocities with the lengths of the transport microtubules, suggesting that the absolute number of motors in the overlaps between template and transport microtubules alone does not influence helical pitch or velocity. Because it was difficult to systematically investigate the influence of the motor density in 3D sliding motility assays with

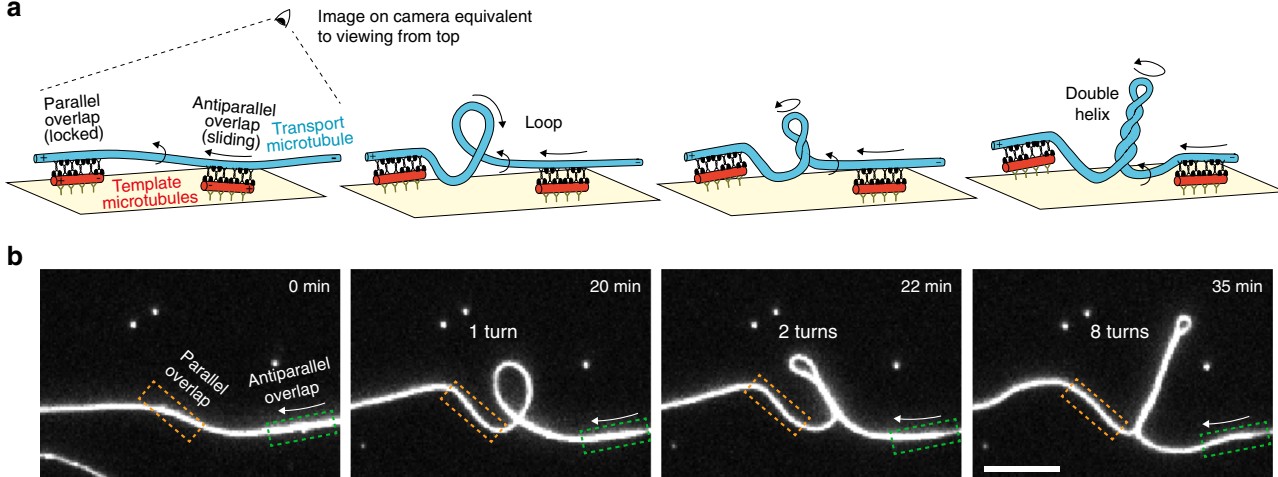

**Fig. 5 Twisting and coiling of Ncd-driven, sliding transport microtubules. a** Schematics of a 2D sliding motility assay on an unstructured glass surface where the leading end of a long transport microtubule (Atto647n-labeled, blue) is locked in a parallel configuration on one template microtubule (rhodamine-labeled, red) while the trailing end of the same transport microtubule is sliding in an antiparallel configuration on another template microtubule. The rotational motion of the transport microtubule twists the central part of the transport microtubule and further causes the transport microtubule to bend into a loop, before coiling up into a double helix. **b** Time-lapse micrographs of the transport microtubule from an event as illustrated in (**a**). The positions of parallel and antiparallel microtubule overlaps (obtained from dual-color micrographs, see Supplementary Movie 4) are indicated by the orange and green boxes, respectively. Scale bar: 10 μm. Three more example events are shown in Supplementary Fig. 4.

suspended microtubules, we employed FLIC-based 2D gliding and sliding motility assays. We found that an increase in the Ncd motor density decreased the microtubule velocity (Supplementary Fig. 4a, b). This finding is in agreement with previous studies on kinesin-14 homologs (HSET[24] and XCTK2[25]) where a similar decrease in the sliding velocity was attributed to steric (longitudinal) hindrance between the motors[24]. At the same time, we observed that the rotational frequency stayed constant upon variation of the motor density (Supplementary Fig. 4a, b) with only slight impediment at very high motor densities. These findings suggest that the collective activity of Ncd motors influences the longitudinal motion (manifested in the sliding velocity) to a larger degree than the rotational motion (manifested by the rotational frequency) and provides a possible explanation for the resulting reduction in the rotational pitch (defined as the ratio of sliding velocity divided by rotational frequency) with increasing motor density. Interestingly, we also observe that the variation in the rotational pitch at a given motor density was significantly lower in the 2D gliding and sliding assays (Fig. 4b, d) compared to the variation in the helical pitch in the 3D sliding assays (Supplementary Fig. 1d). We speculate that the increased viscous drag encountered by transport microtubules moving helically around the template microtubules in the 3D geometry—compared to transport microtubules only rotating around their own axis in 2D assays—make the helical motion significantly more sensitive to the motor density. Previous studies have shown that the viscous drag of individual microtubules can indeed have a significant influence on the motion of microtubules when propelled by loosely anchored motors[18,26]. Taken together, we regard the motor density to be a major contributory factor to the observed variations in the rotational and helical pitches. In fact, while a given concentration of motors is flushed into the sample volume, the effective motor density in a respective microtubule overlap in the 3D assay during sliding may vary depending (i) on the total number and length of template microtubules bound to the surfaces, (ii) on the total number and length of transport microtubules in solution, as well as (iii) on the history of events (e.g., potential accumulation of motors in overlaps where a transport microtubule has already

been sliding on a template microtubule for a significant amount of time).

While we showed that Ncd motors in the sliding geometry cause an intricate axial motion of short transport microtubules, it is not evident that the torque generated by the motors is large enough to be important for intracellular, mechanical events. Both on structured and unstructured glass surfaces (Supplementary Fig. 2b), we observed that the axial motion of transport microtubules was erratic at low Ncd concentrations but became robust at higher Ncd concentrations. This indicates that the magnitude of torque builds up with increasing density of Ncd motors in microtubule overlaps. Furthermore, we demonstrated that long transport microtubules, driven by Ncd motors in an antiparallel microtubule overlap, can cause microtubule twisting and coiling (Fig. 5, Supplementary Movie 4, Supplementary Fig. 5). Similar deformations of cytoskeletal filaments have previously been observed and analyzed in gliding motility assays with dynein and myosin motors[27,28]. In our case, the observed microtubule twisting and coiling is a manifestation of a right-handed torsion generated by Ncd in antiparallel microtubule overlaps, indicating that the motors can induce significant torque which twists torsionally stiff microtubules. In a recent study[18] it was shown that, due to their diffusive tail anchorage, Ncd motors can only generate sub-pN longitudinal forces between overlapping microtubules. However, it is conceivable that Ncd motors are less diffusive in the axial direction, i.e., their sideways diffusivity between neighboring protofilaments might be lower than their diffusivity along a given protofilament. Even rather weak longitudinal motors, such as Ncd, may thus be able to generate significant torques. It will be interesting to further explore the generated torques quantitatively, both for kinesin-14, as well as for other presumably stronger cross-linking motors (like kinesin-5[7,13] and kinesin-8[8,29]). This may be achieved by numerically analyzing the dynamics of the microtubule shapes in assays similar to the experiments presented here or by extending our assay to enable direct torque measurements, for example by using 3D optical tweezers[15]. Most importantly, based on our observations we conjecture that Ncd can generate torques, likely high enough to be relevant in vivo, for example in the mitotic spindle.

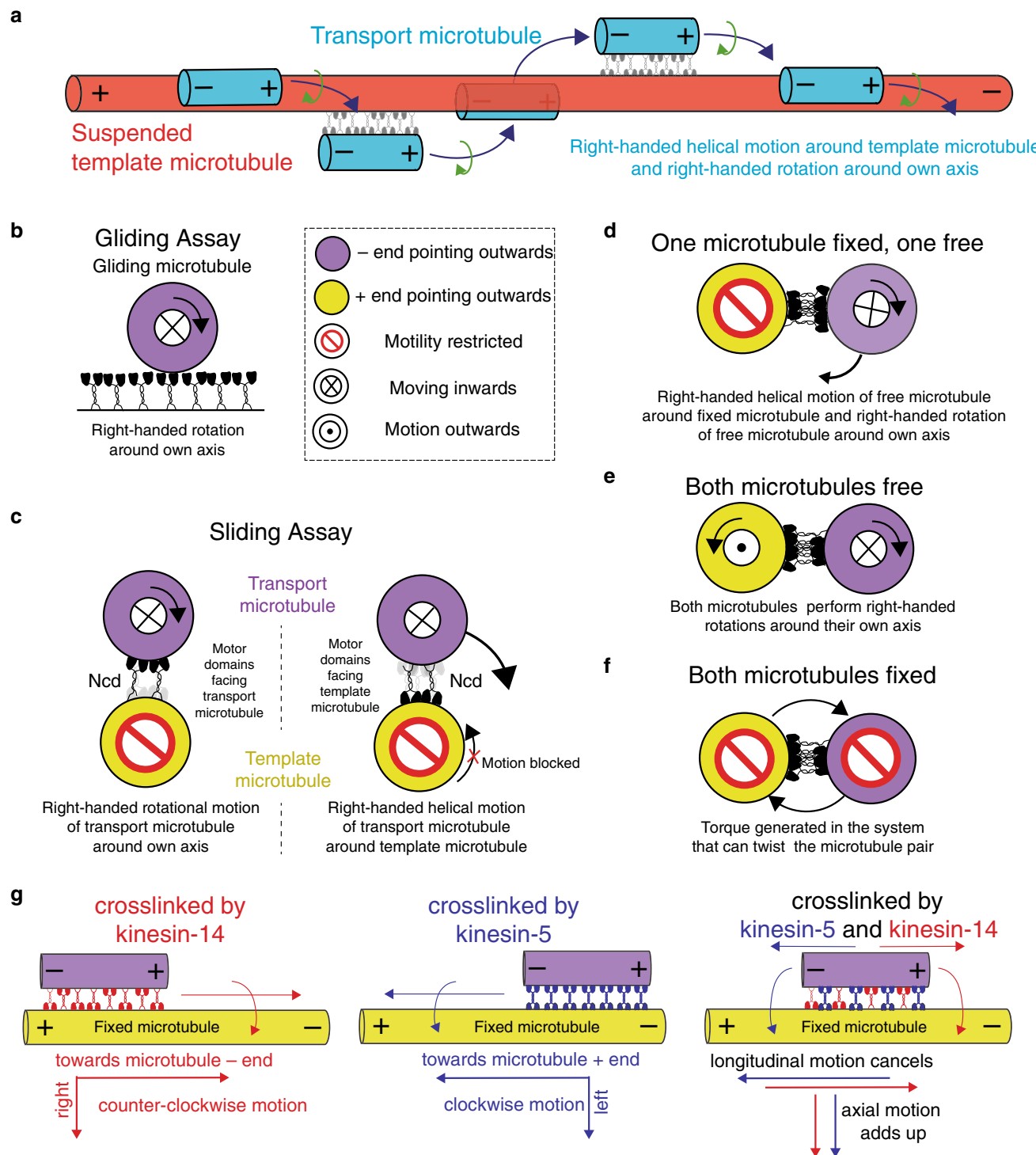

In the mitotic spindle, Ncd cross-linked, antiparallel microtubules can exhibit three geometries: (i) one microtubules is fixed at some point (e.g., attached to the kinetochore[30]) while the other one is free (Fig. 6d). This geometry is similar to what we explore in our 3D sliding motility assays. The free microtubule would perform a right-handed helical motion towards the plus-end of the fixed microtubule, simultaneously rotating around its own axis. (ii) Both microtubules are free (Fig. 6e). In such a situation, both microtubules would perform right-handed rotations around their own axis, and would consequently roll around each other, while sliding apart. (iii) Both microtubules are fixed (e.g., in the stable microtubule overlap formed in the midzone of the mitotic spindle, Fig. 6f). In this geometry, the microtubules would coil around each other and twist the microtubule overlap. In a recent study it was shown that mitotic spindles in HeLa cells are indeed chiral, likely due to torques generated by cross-linking motor proteins[3].

In addition to resolving the 3D trajectories of microtubules sliding around each other, our experimental approach provides a compelling means to measure the extension of motors in their active state while cross-linking and sliding microtubules. For Ncd, we find an in situ extension of 18–21 nm which is in the same range as estimated in earlier electron-microscopy studies for microtubule cross-linker mixtures in yeast cells, including the

**Fig. 6 Illustrations to visualize the motion of antiparallel microtubules cross-linked by Ncd. a** On a suspended template microtubule, antiparallel transport microtubules propelled by Ncd motors exhibit a right-handed helical motion around the template microtubules and a right-handed rotation around their own axis. **b, c** Transverse sections of microtubules propelled by Ncd in the geometries explored in this work. **b** In gliding motility assays, microtubules rotate in a right-handed manner while gliding forward. **c** In sliding motility assays, the Ncd motors bound via their tails to the template microtubule rotate the transport microtubule around its own axis (right-handed), similar to the gliding motility assays. The Ncd motors bound via their tails to the transport microtubule cannot rotate the surface-immobilized template microtubule. This blocked motion is transposed to the transport microtubule, which helically moves around the template microtubule in a right-handed manner. **d–f** Transverse sections of potential rotational motion and torque generation by microtubules cross-linked by Ncd in vivo. **d** If one microtubule is fixed and one microtubule is free, the free microtubule will move similar to what is observed for transport microtubules in our sliding motility assays on suspended template microtubules. The free microtubule will exhibit a right-handed helical motion around the template microtubule and a right-handed rotation around its own axis. **e** If both microtubules are free, they will rotate around their own axis while sliding. **f** If both microtubules are fixed, they will coil around each other or twist their shape depending on how the microtubules are hinged and how strong the torque generated by the motors is. **g** While kinesin-14 motors drive the helical motion of transport microtubules towards the minus-end of the template microtubule in a right-handed manner (counter-clockwise motion), kinesin-5 motors drive the helical motion of transport microtubules towards the plus-end of the template microtubule in a left-handed manner (counter-clockwise motion). Therefore, when kinesin-5 and kinesin-14 are acting together in a microtubule overlap, the motion in the longitudinal direction cancels out while the motion in the axial direction adds up.

homolog kinesin-14, klp2[31]. Interestingly, this distance is lower than estimated for passive cross-linkers from the MAP65 family, with protein extensions ranging between 25 and 35 nm[31–34]. This raises the intriguing question of the actual distance between cross-linked, sliding microtubules in the presence of multiple motor and non-motor cross-linkers. Is it possible that cross-linking motors exhibit different conformation, activity and interaction kinetics depending on how far they are extended? Provided that different microtubule cross-linkers have different natural extensions[31], regulating the microtubule–microtubule spacing might be a smart means of controlling the activity of cross-linking motors and spatially sorting MAPs. Such sorting behavior has previously been reported for the actin bundling proteins, Fascin and $\alpha$-Actinin[35]. Our approach is ideal to measure the extensions (and changes thereof) of different cross-linking motors and MAPs in their active states—in correlation with their functional behavior as generators of longitudinal as well as axial motility and force.

In summary, this study elucidates that Ncd motors induce a helical motion of antiparallelly cross-linked microtubules around each other as well as a rotational motion around their own axis. In vivo, such motion (as opposed to a strict linear motion) might be useful to circumnavigate obstacles on microtubule lattices or in the surrounding environment. While minus-end directed kinesin-14 induces a right-handed rotational motion, other plus-end directed cross-linking kinesins, like kinesin-5, have been shown to induce left-handed rotational motion[7]. In microtubule overlaps where kinesin-5 and kinesin-14 co-exist, the two motors antagonize each other in the longitudinal direction[25] but the torques generated by them are expected to add up (Fig. 6g). Therefore, we hypothesize that in the midzone of mitotic spindles, where stable antiparallel microtubule overlaps are maintained for long periods, cross-linking kinesins build up torques in the system, thereby twisting and possibly coiling microtubules around each other. Finally, the mitotic spindle architecture appears to have an inherent left-handed (moving from spindle center to spindle pole) chirality[4–6]. It is interesting to ponder if this chirality is just an evolutionary artifact that arises from the asymmetry in the power strokes (or step cycles) of the evolutionarily conserved motor domains of kinesin motors or if there is any particular function associated with it.

## Methods

**Protein purification.** Most experiments were performed with recombinant His$_6$-tagged *D. melanogaster* full length GFP-Ncd expressed in SF9 insect cells (cell line IPLB-Sf-21-AE) using a baculovirus expression system[36]. Cells were lysed in 25 mM Tris, 300 mM NaCl, 5 mM imidazole, 5 mM MgCl$_2$, 0.2% (v/v) Tween-20, 10% (v/v) glycerol, 1× protease inhibitor cocktail, 10 mM dithiothreitol (DTT), 1 mM ATP, pH 7.4 and proteins were bound to Ni-NTA resin. Proteins were eluted

by cleavage of the His$_6$-tag with His-tagged PreScission protease[18]. For FLIC-based sliding motility assays, a different batch of full length GFP-Ncd expressed in *Escherichia coli* a was used[12]. Cells were lysed (20 mM Hepes, 1 mM MgCl$_2$, 20 mM 2-mercaptoethanol, 5 μM ADP, 0.1% (v/v) Tween-20, 300 mM NaCl, 20 mM imidazole, 1× protease inhibitor cocktail, pH 7.2 and proteins were bound to a Talon cobalt-affinity resin. After elution with 300 mM imidazole, proteins were further purified with size exclusion chromatography.

**Fabrication of polymer structures on glass.** Cleaned 22 × 22 mm$^2$ glass coverslips (#1.5; Menzel, Braunschweig, Germany) were imprinted with a UV curable resin, EVG NIL UV/A 200 nm (EV Group) using UV nanoimprint lithography (UV-NIL) as described in Mitra et al.[9]. The structure imprinted on the glass coverslips was characterized by repeated pattern of relief lines (that form the ridges) with a height of 370 nm (few experiments were also performed on 250 nm high ridges) and a width of 2 μm (or 5 μm), separated by 10 μm wide valleys between the ridges.

**Microtubule preparation.** Both, template and transport microtubules, were gua-nylyl-($\alpha,\beta$)-methylene-diphosphonate (GMP-CPP) grown, taxol-stabilized (refer-red to as double stabilized). Template microtubules were long (average length > 15 μm) and rhodamine-labeled, while transport microtubules were short (average length 1–2 μm) and Atto647n-labeled. Totally, 4.6 μM rhodamine-labeled *porcine* tubulin was added to a polymerization solution, comprising of BRB80 (80 mM Pipes at pH 6.9, 1 mM MgCl$_2$, 1 mM EGTA) supplemented with 1 mM GMP-CPP [Jena Bioscience, Jena, Germany] and 4 mM MgCl$_2$, incubated on ice for 5 min followed by 30 min at 37 °C, to grow short microtubule seeds. The solution was centrifuged at 17,000×g for 15 min at 25 °C to remove free tubulin and the pellet was resuspended in a new polymerization solution supplemented with 0.4 μM of rhodamine-labeled tubulin. This solution was incubated overnight at 37 °C with low tubulin concentration allowing microtubule seeds to anneal and form long microtubules. The solution was then centrifuged at 17,000×g for 15 min at 25 °C and the pellet was resuspended in BRB80T solution (BRB80 supplemented with 10 μM taxol). Transport microtubules were grown as short Atto647n-labeled microtubule seeds in the same way as described for the first cycle of polymerization for template microtubules.

For the FLIC-based sliding motility assays, Cy5-labeled template microtubules (grown as described above) and rhodamine-speckled transport microtubule (grown as described in Mitra et al.[17]) were used. For the FLIC-based gliding motility assays, rhodamine-speckled microtubules were used.

For the microtubule coiling assays, taxol-stabilized Atto647n-labeled transport microtubules were grown at 37 °C for 2.5 h in BRB80 supplemented with 30 μM Atto647n-labeled tubulin, 4.8% (v/v) dimethyl sulfoxide, 4 mM MgCl$_2$ and 1 mM Guanosine-5′-triphosphate (GTP). Microtubules were sedimented as described above, resuspended in BRB80T and allowed to anneal for 48 h.

**Ncd-driven microtubule sliding motility assays.** Motility buffer (MB), used in all motility assays, consisted of 20 mM Hepes at pH 7.2, 1 mM EGTA, 2 mM MgCl$_2$, 75 mM KCl, 10 μM taxol, 200 μg mL$^{-1}$ casein, 10 mM DTT, 0.1% (v/v) Tween-20, 20 mM D-glucose, 100 μg mL$^{-1}$ glucose oxidase, 10 μg mL$^{-1}$ catalase and either 1 mM ATP (MB-ATP) or 1 mM ADP (MB-ADP). 3D sliding motility assays on suspended template microtubule in microfluidic flow cells con-structed on 22 × 22 mm$^2$ glass coverslips patterned with UV-NIL polymer resin and 18 × 18 mm$^2$ unpatterned glass coverslips, both dichlorodimethylsilane (DDS)-coated to make the surface hydrophobic[37]. Before silanization, the patterned coverslips were cleaned mildly (using 5% mucasol and then 70% ethanol) to avoid corrosion of the structure. 2D sliding motility assays on surface-immobilized microtubules were performed on unpatterned silanized coverslips or silicon wafers (10 × 10 mm$^2$) with a 30 nm thermally grown oxide layer (for FLIC based motility

assays; GESIM, Grosserkmannsdorf, Germany). In both, 2D and 3D sliding assays, flow cells were flushed with the following sequence of solutions: (i) Bead solution consisting of 2% (v/v) 200 nm Tetraspeck beads (incubation time 1 min; ThermoFisher Scientific). (ii) Antibody solution consisting of 20–200 μg mL$^{-1}$ anti-rhodamine antibody (Mouse monoclonal clone 5G5; ThermoFisher Scientific) in phosphate-buffered saline (PBS) for unspecific binding of antibodies to the surface (incubation time 5 min). (iii) 1% pluronic F-127 in PBS (Sigma) in order to block the surface from unspecific protein adsorption (incubation time >60 min). (iv) BRB80 washing step to remove unbound F-127 and exchange buffers. (v) Rhodamine-labeled template microtubule solution in BRB80T, followed by an immediate washing step with MB, in order to immobilize microtubules perpendicular to the ridges. (vi) MB-ADP solution containing Ncd (concentration ranging between 0.02 and 40 nM) for the motors to bind to the template microtubules. (vii) MB-ADP solution containing Atto647n-labeled transport microtubules, followed by immediate washing step with MB-ADP, in order to cross-link few transport microtubules to the template microtubules and wash away the unbound ones. (viii) MB-ATP solution at the microscope after finding a suitable field of view. For FLIC-based sliding motility assays, Cy5-labeled template microtubules were immobilized on the surface using anti-Cy5 antibodies (Mouse monoclonal clone CY5-15; Sigma-Aldrich; working concentration 75 μg/mL) and rhodamine-speckled transport microtubules were used.

**Ncd-driven microtubule gliding motility assays.** For 2D FLIC-based gliding motility assays on Fab-fragment (anti-mouse IgG [Fc specific]; polyclonal; Sigma-Aldrich, working concentrations 0.04–1 mg/mL) and mouse anti-GFP antibody-coated surfaces (Mouse monoclonal clone 106A20; in-house Protein Facility, Max Planck Institute for Molecular Cell Biology and Genetics, Dresden, Germany; working concentration 0.15 mg/mL), the assay was performed in flow cells constructed from DDS-coated silicon wafers and glass coverslips as described in Mitra et al.[17] with replacement of BRB80 based motility buffer with MB-ATP.

**Image acquisition.** Optical imaging was performed using an inverted fluorescence microscope (Axio Observer Z1; Carl Zeiss Microscopy GmbH) with a 63× oil immersion 1.46NA objective (Zeiss) in combination with an EMCDD camera (iXon Ultra; Andor Technology) controlled by Metamorph (Molecular Devices Corporation). A LED white light lamp (Sola Light Engine; Lumencor) in combination with a TRITC filterset (ex 520/35, em 585/40, dc 532: all Chroma Technology Corp.) and an Atto647n filterset (ex 628/40, em 692/40, dc 635; all Chroma Technology Corp.), corresponding to rhodamine-labeled microtubules and Atto647n/Cy5-labeled microtubules, respectively, were used for epifluorescence imaging. The imaging temperature was maintained at 24 °C by fitting a custom-made hollow brass ring around the body of the objective and connecting it to a water bath with a cooling/heating unit (F-25-MC Refrigerated/Heating Circulator; JULABO GmbH)[38]. For sliding motility assays, a field of view was selected when there were three (or more) Tetraspeck beads bound to the surface and several trackable template microtubules suspended between ridges. Template microtubules were imaged in the TRITC channel (for 50–100 frames at 3–10 fps with exposure time 100–300 ms) before and after imaging the transport microtubules in the Atto647n channel (for 5–15 min at 3–10 fps with exposure time 100–300 ms) to confirm that the template microtubules do not move while imaging the transport microtubules. For FLIC-based motility assays, a 63× water immersion 1.2NA objective (Zeiss) was used (higher working distance) in order to image rhodamine-speckled microtubule (for 5–15 min at 1 fps in the TRITC channel with exposure time 400 ms) on the silicon wafer surfaces on the far side of the flow cells.

**Sideways distance measurements.** The acquired image streams of immobilized template microtubules (TRITC channel) and sliding transport microtubules (Atto647n channel) were analyzed using FIESTA[16] (version 1.6.0). First, the Tetraspeck beads, that serve as fiducial markers, were tracked in both channels to obtain the color offset correction (nonreflective similarity: allowing for translation, rotation, and scaling) between the two channels as well as the image drift correction in the corresponding channels. Template microtubules with relevant sliding events were tracked. Since template microtubules were immobilized, the filament position (tracked over 50–100 frames) was averaged to obtain the filament position. Sliding transport microtubules (antiparallel events) and locked transport microtubules (parallel events) on template microtubules with sliding events (to know the microtubule orientation) were tracked. After color and drift correction, the sideways distance was obtained as the perpendicular distance of the center point of the tracked transport microtubule from the averaged center line (tracked over the recorded 50–100 frames) corresponding to the template microtubule (illustrated in Fig. 1b). For displaying the sideways distance plots, data were smoothened by rolling frame averaging over 20 consecutive frames.

**Analysis of sliding events on suspended microtubules.** Rotational pitch, end-to-end velocity and the diameter of the helical path corresponding to each rotation of a helical sliding event was determined by manual computer-aided measurement of the sideways distance versus forward distance plots. For a given sliding event, measurements from individual rotations were averaged to obtain the mean pitch, velocity and diameter of helical path. Accounting for the helicity

of the path traversed by the sliding microtubule, the actual velocity along the path (referred to as contour velocity in Supplementary Fig. 1) was calculated assuming a helical path with a diameter of 86 nm

$$\left( \text{Contour Velocity} = \sqrt{(\text{Velocity})^2 + (\text{Velocity}/\text{Pitch} \times 2\pi r)^2}; r = 43\,\text{nm} \right).$$

**Analysis of rotational data in the FLIC motility assays.** The rotational pitch of the gliding or sliding microtubules in the FLIC-based motility assays was obtained from their kymographs, which were generated in Fiji[39] using the MultiKymograph Plugin (version 3.0.1). The kymographs were then analyzed with MATLAB (Mathworks, USA) using the speckle analysis method[17].

**Analysis of distributions with error estimation.** For estimating parameters from any given distribution (e.g., sideways distance, pitch, and diameter of helical path) we used a bootstrapping approach[40,41]. Here, the distribution (N number of measurements) was resampled by randomly picking N measurements from the measured distribution (with replacement) and calculating the median of the resampled distribution. This was repeated 1000 times. The resulting bootstrapping distribution was used to estimate the parameter (mean of the bootstrapping distribution μ) and its error (standard deviation of the bootstrapping distribution σ). All values and errors as well as error bars in this paper use $\mu \pm 3\sigma$ (99% confidence interval), unless otherwise noted.

**Reproducibility.** Data for the microtubule coiling experiments were acquired from five independent experiments. Data for all other experiments were acquired during at least three independent experimental days, performed over several months.

**Data representation.** In boxplots (Fig. 4b, Supplementary Fig. 4) midline indicates the median; the hollow square in the middle indicates the mean; bottom and top box edges indicate the 25th and 75th percentiles, respectively; the whiskers extend to the most extreme data points not considered as outliers; the most extreme data points are indicated by crosses. The individual data points are shown in the background.

## Data availability

Data supporting the findings of this manuscript are available from the corresponding authors upon reasonable request. A reporting summary for this Article is available as a Supplementary Information file.

The source data underlying Figs. 1c, 2a–c, 3a–b, 4a–d, Supplementary Figs. 1a–d, 2b, 3a, b, 4a–b, d are provided as a Source Data file.

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

## Acknowledgements
We thank Andrej Vilfan and Bert Nitzsche for scientific discussions and comments on the paper, all members of the Diez laboratory for fruitful interactions, Salvatore Girardo and the Microstructure Facility at the Center for Molecular and Cellular Bioengineering at TU Dresden for preparing the polymer-structured coverslips and Corina Bräuer for technical support. We acknowledge financial support from the Deutsche Forschungsgemeinschaft through the Sonderforschungsbereich 1027 (Project A8), the Max Planck Institute for Molecular Cell Biology and Genetics Dresden and the Technische Universität Dresden.

## Author contributions
A.M., F.R., and S.D. designed the research; A.M., L.M., R.G., and R.R. performed the research; F.R. contributed the reagents/analytic tools; A.M., L.M., and R.G. analyzed the data; A.M. and S.D. wrote the paper with comments from the other authors.

## Competing interests
The authors declare no competing interests.
