## [Peer Review File · Nature Communications]

Reviewers' comments:

Reviewer #1 (Remarks to the Author):

The authors systematically compare the rotational motion of microtubules that Ncd motors crosslink in vitro. It is well known that several mitotic kinesins such as kinesin-5 and kinesin-14 crosslink two microtubules. An emerging view is that they exhibit off-axis components and generate the rotation of a microtubule around the long axis of the microtubule as they move on the other microtubule, suggesting an important role in generating twisting of the mitotic spindle. Despite its importance of Ncd in the spindle organization, the rotational motion of microtubules driven by Ncd has not been sufficiently addressed so far. The authors presented a discovery that Ncd can generate the coiling of microtubule bundles, leading to the generation of chirality in the mitotic spindle possibly in collaboration with kinesin-5. A weak point of this paper is that the authors were not able to assess the dependence of the motor density on the helical pitches in 3D sliding motility assays. However, this point was carefully addressed by 2D microtubule motility assays performed on a planar glass surface using a FLIC-based technique. This paper will be an important contribution to the biophysics and cell biology fields, and can also be a useful guide for those who are interested in applying the 3D measurement of nanometer-scale objects to their own research fields. The manuscript is well organized and described. I recommend that it be accepted for publication in Nature Communications.

Additional comments:

- The difference between the current study and the previous work (Nitzsche et al., PNAS 2016) seems a bit unclear since both papers describe microtubule gliding on Ncd-coated surfaces that rotated around their longitudinal axes. What is novel in the current paper is the possibility that Ncd can generate the coiling of microtubule bundles, which is relevant to the recent finding about the chirality of the mitotic spindle. Although, in principle, this result could be anticipated from the previous results that Ncd exhibits off-axis components in the microtubule gliding assays, the actual experimental data about crosslinked microtubule bundles are of critical importance when building a molecular model of the mitotic spindle assembly.

- If possible, it might be better to mention briefly the quantitative relationship between the size/direction of the powerstroke of Ncd that described in the previous study, and the resultant rotational motion of microtubules. Although this point might be discussed in the previous study (Nitzsche et al., PNAS 2016), the discussion must be too complicated for readers to follow.

Reviewer #2 (Remarks to the Author):

In this paper Mitra et al. have upgraded an in vitro system developed previously by the same authors (Mitra et al. PNAS 2018) to show that Ncd, a non processive minus end directed motor protein that has been shown to slide microtubules (Fink et al. Nat Cell Biol, 2009), in a three-dimensional setting can carry a short stabilised antiparallel microtubule around a longer one following a helical, right-handed trajectory. Even though this is the first time such a specific setting has been shown, the amount of previously published data that already shows helical motion of different kinesin motors (as cited by authors themselves) and observations that spindle is a helical structure in both human and yeast cells (Novak et al. Nat. Comm 2018; Winey et al. J. Cell Biol 1995; Ding et al. J Cell Biol. 1993) in my opinion do not make this paper in its current state acceptable for publication in Nature Communication. Even though there is a difference between a motor protein carrying a quantum dot and several motor proteins carrying a short microtubule, I was not convinced by the authors' arguments to see how this work represents a significant jump in our current understanding of microtubule sliding, especially in the case of the mammalian mitotic spindle where microtubules of similar lengths seem to coil around each other in order to produce torque of microtubule bundles. Rotation of a short microtubule around bigger one is not the same

as twisting of microtubules around each other.

This being said I believe the work can be accepted for publication with some additional experimental data. As I mentioned the biological motivation of the authors seems to be the mitotic spindle. In that case, as proposed in the cartoon of the Figure 5F, the authors should do an experiment involving two long and fixed microtubules to see whether the motor proteins can exert enough force to cause the coiling of two microtubules. This data would in my opinion be much more informative for the mitotic spindle context, and would actually mirror what is believed to be happening in vivo, where microtubules are fixed at the spindle poles.

Further comments:

The authors on more than one occasion misquote the Novak et al. article as saying the spindle has right-handed chirality. This is not true, as the article clearly states that the spindle has left-handed chirality. In the last section of Discussion, the authors discuss how the interplay of Eg5 and Ncd might produce such chirality of the spindle. Taking into account that the spindle is left-handed and not right-handed, the authors should adjust their arguments and rewrite the discussion accordingly. Statement that Eg5 and Ncd with their movements of opposite handedness and direction result in torques that add up would benefit from a graphical depiction to make it more understandable. This statement in my opinion is rather important even though it is part of discussion, since it tries to explain the possible interplay of motors with different directionalities and torques present in vivo.

Probably my biggest concern with this work is that In Figure 2A the pitches vary considerably with the same concentration at 4nM Ncd. It seems to me they vary much more within this one concentration, than in the Figure 4D where the concentration of the Ncd was changed? Also in Figure 4D there seems to be no difference between 2.4 nM and 6 nM Ncd and no pitch is close to 3 μm , observable at 4nM in Figure 2A. How do authors explain these inconsistencies apart from the fact that in the latter experiment they are using a 2D motility assay and not the suspended microtubule assay?

In a situation with parallel transport microtubules related to Figure 2 and Supplementary Figure 2, the comparison between suspended template (Figure 2C) and surface-immobilized template (Supplementary Figure 2C) is confusing. The distribution on the surface-immobilized template in my opinion does not show a "minor bias to the right" as stated by authors but shows a quite big bias to the right. In any case, for me this difference to the bimodal distribution seen in suspended template seems unexpected, why the authors think this is happening?

Also, a clearer way of quantifying this bias would help to avoid imprecise wording.

The authors attribute to autofluorescence the inability to measure the concentration of Ncd-GFP. Would it be possible to measure it with another fluorescent tag? Would a time-gated imaging approach be helpful to eliminate autofluorescence in this setup?

In supplementary figure 4C caption text, I am unconvinced by the hard claim that "At 0.6nM Ncd the microtubule did not show any clear rotational movement". Looking at the example trajectory shown below the kymographs it is clearly visible that at 0.6 nM there is rotational movement. To me it looks quite similar to what is observed after the switch of concentration considering its much shorter duration. Moreover, the authors quantify this rotational movement at 0.6 nM Ncd as having the rotational frequency similar to 6 nM Ncd (supplementary figure 4B right panel). Also, in Figure 4 at 0.6 nM Ncd, the rotational pitch is close to 2 whereas at 6 nM Ncd it is close to 0.8. In the main text however the authors are more prudent with their wording and talk about "few transport microtubules" that rotate at 0.6 nM.

I noticed in these experiments the very low N for 0.6 nM Ncd experiments (N=7) compared to N=29 for 6 nM Ncd. Why is this the case? On the same note both Figure 4 and Supplementary Figure 4 lack p-values.

The mention of chirality observed in yeast spindles in my opinion also merits a mention in the discussion, specifically Winey et al. J. Cell Biol 1995; Ding et al. J Cell Biol. 1993.

Supplementary Movie 3 is not referenced in the main text.

Supplementary Figure 4E is referenced in the text, yet there is no Supplementary Figure 4E.

We would like to thank the reviewers for their constructive comments on our work. We have addressed the issues raised by the reviewers and altered the manuscript accordingly.

Most importantly, we (i) added new experimental data on the actual generation of torque in microtubule-microtubule overlaps, (ii) extended our discussion with regard to the variations in the helical pitches, and (iii) more clearly relate our results to (our) earlier work.

Below, please, find the point-by-point reply to all the raised comments.

Reviewer #1 (Remarks to the Author):

The authors systematically compare the rotational motion of microtubules that Ncd motors crosslink in vitro. It is well known that several mitotic kinesins such as kinesin-5 and kinesin-14 crosslink two microtubules. An emerging view is that they exhibit off-axis components and generate the rotation of a microtubule around the long axis of the microtubule as they move on the other microtubule, suggesting an important role in generating twisting of the mitotic spindle. Despite its importance of Ncd in the spindle organization, the rotational motion of microtubules driven by Ncd has not been sufficiently addressed so far. The authors presented a discovery that Ncd can generate the coiling of microtubule bundles, leading to the generation of chirality in the mitotic spindle possibly in collaboration with kinesin-5. A weak point of this paper is that the authors were not able to assess the dependence of the motor density on the helical pitches in 3D sliding motility assays. However, this point was carefully addressed by 2D microtubule motility assays performed on a planar glass surface using a FLIC-based technique. This paper will be an important contribution to the biophysics and cell biology fields, and can also be a useful guide for those who are interested in applying the 3D measurement of nanometer-scale objects to their own research fields. The manuscript is well organized and described. I recommend that it be accepted for publication in Nature Communications.

Response: We would like to thank the reviewer for her/his positive review and useful comments.

With regard to the influence of the motor density on the helical pitches, we indeed found it difficult to address the issue in our 3D sliding assays on suspended microtubules (due to the reasons already elaborated in our original submission). Nevertheless, we were able to study the dependence of motor density on the rotational pitches in 2D gliding/sliding assays and believe that these might partially explain the variation in helical pitches in the 3D assay. However, there are likely other factors that contribute to the large variation of helical pitches. We now discuss this in more detail in the discussion of our manuscript (see also our response to Reviewer 2 below).

Additional comments:

The difference between the current study and the previous work (Nitzsche et al., PNAS 2016) seems a bit unclear since both papers describe microtubule gliding on Ncd-coated surfaces that rotated around their longitudinal axes. What is novel in the current paper is the possibility that Ncd can generate the coiling of microtubule bundles, which is relevant to the recent finding about the chirality of the mitotic spindle. Although, in principle, this result could be anticipated from the previous results that Ncd exhibits off-axis components in the microtubule gliding assays, the actual experimental data about crosslinked microtubule bundles are of critical importance when building a molecular model of the mitotic spindle assembly.

Response: The crucial difference between Nitzsche et al. (PNAS, 2016) and our current work is the geometry in which the motors perform their powerstrokes. Nitzsche et al., and a couple of other earlier studies, show that Ncd motors bound rigidly to a solid/planar surface (conventional surface gliding assays) can rotate microtubules. In our current study, we demonstrate that in the microtubule-microtubule sliding geometry (where the microtubules are linked to each other by Ncd motors diffusively coupled to one microtubule via their tail domains and interacting with the other microtubule by their motor domains), Ncd can rotate “transport” microtubules around their own longitudinal axis as well as move them helically around the “template” microtubules. While this might appear to be a logical extension of Nitzsche et al., we would like to note that the force generation of Ncd is quite different for rigid vs. diffusive coupling (see Luedecke et al., Nat. Comm., 2017).

Action taken: We now highlight this difference more clearly in the first paragraph of the Discussion.

Additional Response: Moreover, we now directly demonstrate the generation of torque that leads to microtubule twisting and coiling (new data in Fig. 5, Supplementary Figure 5 and Supplementary Movie 4). We believe, these novel results considerably substantiate our claim that Ncd motors, in the physiologically relevant microtubule-microtubule sliding geometry, can generate enough torque to twist microtubules, which can provide an inherent chirality to the mitotic spindle. At the same time, we regard this finding a significant advance over what could be expected from Nitzsche et al..

If possible, it might be better to mention briefly the quantitative relationship between the size/direction of the powerstroke of Ncd that described in the previous study, and the resultant rotational motion of microtubules. Although this point might be discussed in the previous study (Nitzsche et al., PNAS 2016), the discussion must be too complicated for readers to follow.

Reply: Good idea.

Action taken: Referring to Nitzsche et al. (PNAS 2016) we now state in the Discussion that the helical pitches measured in our 3D microtubule-microtubule sliding assays are consistent with the rotational pitches observed for microtubules gliding on surface-bound Ncd motors and also specify the longitudinal stalk displacement of 9 nm accompanied by a net angular displacement of 1.1° of the individual motors.

Reviewer #2 (Remarks to the Author):

In this paper Mitra et al. have upgraded an in vitro system developed previously by the same authors (Mitra et al. PNAS 2018) to show that Ncd, a non processive minus end directed motor protein that has been shown to slide microtubules (Fink et al. Nat Cell Biol, 2009), in a three-dimensional setting can carry a short stabilized antiparallel microtubule around a longer one following a helical, right-handed trajectory. Even though this is the first time such a specific setting has been shown, the amount of previously published data that already shows helical motion of different kinesin motors (as cited by authors themselves) and observations that spindle is a helical structure in both human and yeast cells (Novak et al. Nat. Comm 2018; Winey et al. J. Cell Biol 1995; Ding et al. J Cell Biol. 1993) in my opinion do not make this paper in its current state acceptable for publication in Nature Communication. Even though there is a difference between a motor protein carrying a quantum dot and several motor proteins carrying a short microtubule, I was not convinced by the authors' arguments to see how this work represents a significant jump in our current understanding of microtubule sliding, especially in the case of the mammalian mitotic spindle where microtubules of similar lengths seem to coil around each other in order to produce torque of microtubule bundles. Rotation of a short microtubule around bigger one is not the same as twisting of microtubules around each other.

Response: We thank the reviewer for the positive remarks and the constructive assessment of our novel assay, showing the 3D motion of two microtubules sliding around each other.

With regard to comparing our assay geometry to Mitra et al. (PNAS 2018) where quantum dots, rigidly attached to single, processive kinesin-8 motors, were shown to helically move around suspended template microtubules we would like to highlight a number of crucial differences. In our current assay (i) the transport (cargo) microtubules are only diffusively attached to the motors via slippery tail interactions (potentially leading to a significant breakdown in force, see Luedecke et al., Nat. Comm., 2017), (ii) the motors are oriented bidirectionally between the template and transport (cargo) microtubules, i.e. there are also motors facing the transport (cargo) microtubule, and (iii) the assay geometry is even more intricate as the transport (cargo) microtubules additionally rotate around their own axes.

Action taken: We now highlight these differences more clearly in the text, e.g. by stating in the Introduction that we track the 3D motion of "crosslinked" short microtubules ... rather than the "transport" of these microtubules.

Response: With respect to the lengths of the transport (cargo) microtubules we note that studied lengths are in the range of 600-2500 nm. Recent 3D reconstructions of mitotic spindles (in *C. elegans* and human cells) show that most spindle microtubules are less than 2000 nm (see Figure 3B, corresponding to the mitotic spindle of a *C. elegans* embryo from Redemann et al., Nat. Comm., 2017.). Therefore, we believe that the microtubule lengths we studied are similar to the physiologically relevant length range.

Nevertheless, we agree with the reviewer, that “Rotation of a short microtubule around another one is not the same as twisting of microtubules around each other.”

Action taken: To address the latter point, we added new data to our manuscript where we clearly show that the rotational motion of a transport microtubule sliding on an immobilized template microtubule generates enough torque to cause microtubule twisting and coiling (new Fig. 5, Supplementary Figure 5 and Movie 4, see also our detailed response below).

This being said I believe the work can be accepted for publication with some additional experimental data. As I mentioned the biological motivation of the authors seems to be the mitotic spindle. In that case, as proposed in the cartoon of the Figure 5F, the authors should do an experiment involving two long and fixed microtubules to see whether the motor proteins can exert enough force to cause the coiling of two microtubules. This data would in my opinion be much more informative for the mitotic spindle context, and would actually mirror what is believed to be happening *in vivo*, where microtubules are fixed at the spindle poles.

Response: Yes, we fully agree and it has indeed been our plan to study this carefully in potential follow up work. Nevertheless, we do understand the importance of providing evidence that the microtubule rotations in the microtubule-microtubule sliding geometry can lead to microtubule twisting and are happy to report that we eventually managed to show exactly this.

The reviewer suggests to directly test if Ncd motors can twist or coil two partially fixed microtubules, at an antiparallel microtubule overlap, in order to assure that the axial motion induced by Ncd is biologically relevant. We thoroughly attempted to recreate the suggested experimental geometry but faced several technical challenges. We therefore resorted to an alternative geometry, which in our opinion shows the very same. We performed sliding motility assays on unstructured glass surfaces with long transport microtubules and short, surface-immobilized template microtubules. We focused on events, where the leading end of a transport microtubule was locked in a parallel configuration on one template microtubule and the trailing end of the same transport microtubule was sliding in an antiparallel configuration on another template microtubule. In multiple experiments we found that the central parts of those transport microtubules were often twisted into loops before being coiled up into double-helices with up to eight turns. This indicates that the right-handed torque generated by Ncd motors in the sliding geometry is high enough to twist and coil microtubules. Based on these observations, we conjecture that Ncd can generate torques, likely high enough to be of importance *in vivo*, for example in the mitotic spindle.

We are aware that we only present qualitative evidence of microtubule coiling. We will aim to quantify the involved forces/torques in our future work but regard this beyond the scope of the current manuscript.

Action taken: We now present and discuss the above-mentioned new results in the new Fig. 5, Supplementary Figure 5 and Supplementary Movie 4.

Further comments:

The authors on more than one occasion misquote the Novak et al. article as saying the spindle has right-handed chirality. This is not true, as the article clearly states that the spindle has left-handed chirality. In the last section of Discussion, the authors discuss how the interplay of Eg5 and Ncd might produce such chirality of the spindle. Taking into account that the spindle is left-handed and not right-handed, the authors should adjust their arguments and rewrite the discussion accordingly.

Response: We apologize for this confusion. In Novak et al., it is indeed found that when one looks at the mitotic spindle from the spindle poles the bundles are arranged in a clockwise manner. Consequently, the chirality of the spindle is left-handed if you move **from the spindle center to spindle pole** ... consistent with what we wrote in the discussion “*the mitotic spindle architecture appears to have an inherent right-handed (moving from spindle poles to center) chirality*”.

Action taken: For clarity we have now changed that statement in accordance with the description in Novak et al. at all relevant locations in our manuscript.

Statement that Eg5 and Ncd with their movements of opposite handedness and direction result in torques that add up would benefit from a graphical depiction to make it more understandable. This statement in my opinion is rather important even though it is part of discussion, since it tries to explain the possible interplay of motors with different directionalities and torques present in vivo.

Response: Very good idea.

Action taken: We added a new panel (G) to Figure 6.

Probably my biggest concern with this work is that in Figure 2A the pitches vary considerably with the same concentration at 4nM Ncd. It seems to me they vary much more within this one concentration, than in the Figure 4D where the concentration of the Ncd was changed? Also in Figure 4D there seems to be no difference between 2.4 nM and 6 nM Ncd and no pitch is close to 3 μm , observable at 4nM in Figure 2A. How do authors explain these inconsistencies apart from the fact that in the latter experiment they are using a 2D motility assay and not the suspended microtubule assay?

Response: This is a good question and we have given a lot of thought to the variations in rotational pitches observed between the events in the 3D assay (Figure 2A). We cannot give a definite answer but strongly believe that the actual motor density in the respective overlaps plays an important role.

The results of the 2D assays shown in Figure 4 (sliding and gliding) are hinting that the density of motors is likely a contributory factor to the variation in the rotational and helical pitches. In fact, while 4 nM Ncd is the concentration that we flush into the channel in the 3D assay, the effective density of motors in the respective microtubule overlaps may vary depending (i) on the total number and lengths of template microtubules bound to the channel surface, (ii) on the total number and length of transport microtubules in solution, as well as (iii) on the history of events (e.g. potential accumulation of motors in overlaps where a transport microtubule has already been sliding on a template microtubule for a significant amount of time). These possibilities may also explain why we even observe that the helical pitches do vary substantially for transport microtubules sliding on the same template microtubule.

Further, we speculate, that small variation in motor density may translate into larger variations in the helical pitches than in the rotational pitches due to the higher viscous drag of the helically moving microtubules in the 3D geometry (as they not only move forward but also sideways). In fact, we know that the viscous drag already does significantly influence the forward speed of microtubules when propelled by loosely anchored motors (Grover et al., PNAS, 2016 and Lueddecke et al., Nat. Comm., 2017).

The fact that in Figure 4D, there is not a big difference in rotational pitch (though significant; see Supplementary Figure 4C) between 2.4 nM and 6 nM Ncd could be due to the rather low difference in the effective motor densities in the two experiments. As can be seen in the Supplementary Figure 4B, the velocities are also very similar at the 2 concentrations.

Action taken: We now explain our line of thoughts more extensively in the discussion.

In a situation with parallel transport microtubules related to Figure 2 and Supplementary Figure 2, the comparison between suspended template (Figure 2C) and surface-immobilized template (Supplementary Figure 2C) is confusing. The distribution on the surface-immobilized template in my opinion does not show a "minor bias to the right" as stated by authors but shows a quite big bias to the right. In any case, for me this difference to the bimodal distribution seen in suspended template seems unexpected, why the authors think this is happening? Also, a clearer way of quantifying this bias would help to avoid imprecise wording.

Response: That is a good catch! We believe, two factors might be causing this discrepancy. (i) When a transport microtubule lands on a template microtubule, the motors are not oriented equally in both directions (i.e. there might be different motor densities on template and transport microtubules) in ADP but equilibrate upon addition of ATP (see also Fink et al. Nat Cell Biol, 2009). During this time of equilibration, the parallel microtubules can still slide along each other to some extent (as the directional motor forces do not fully cancel each other), moving forward and to the right, which provides the bias that we observe. (ii) Occasionally we observe the flipping of a transport microtubule at the minus-end of a template microtubule. It is likely that these formerly-moving (and thus right-side oriented) microtubules become parallel and locked to the right side of the template microtubule upon flipping.

Action taken: In our original submission we did show Supplementary Figure 2C for completeness when comparing the 2D/3D events and because the peak was hinting towards the same distance as in Supplementary Figure 2B. However, due to the complexity and the questions that may arise (and which are not the focus of the paper), we decided to remove this panel and the corresponding text.

The authors attribute to autofluorescence the inability to measure the concentration of Ncd-GFP. Would it be possible to measure it with another fluorescent tag? Would a time-gated imaging approach be helpful to eliminate autofluorescence in this setup?

Response: These are very good ideas and we will keep them in mind for potential future experiments using the patterned surfaces. Likewise, we may also consider pre-bleaching the patterned polymer structures. Quantifying the number and/or density of motors in interaction with cytoskeletal filaments is an issue of great importance always coming up again. Thus, technology development into this direction may be worth doing.

In our present case, we were however limited in the choice of fluorophore, considering that both template and transport microtubules need to be tracked with high accuracy for our measurements. If Ncd is tagged with a far-red dye it should be possible to measure the intensity of motors. However, it would then not be possible to track microtubules with high resolution. The autofluorescence is very high in the GFP range, lower in the rhodamine range and not present in the Atto647 range (common for UV cured polymers). Thus we image the transport microtubules in the Atto647 channel.

In supplementary figure 4C caption text, I am unconvinced by the hard claim that “At 0.6nM Ncd the microtubule did not show any clear rotational movement”. Looking at the example trajectory shown below the kymographs it is clearly visible that at 0.6 nM there is rotational movement. To me it looks quite similar to what is observed after the switch of concentration considering its much shorter duration.

Moreover, the authors quantify this rotational movement at 0.6 nM Ncd as having the rotational frequency similar to 6 nM Ncd (supplementary figure 4B right panel). Also, in Figure 4 at 0.6 nM Ncd, the rotational pitch is close to 2 whereas at 6 nM Ncd it is close to 0.8. In the main text however the authors are more prudent with their wording and talk about “few transport microtubules” that rotate at 0.6 nM.

I noticed in these experiments the very low N for 0.6 nM Ncd experiments (N=7) compared to N=29 for 6 nM Ncd. Why is this the case?

Response: The rotational pitches of the speckled microtubules were obtained using the speckle analysis method described in Mitra et al. 2015, PlosOne. Briefly, the rotational pitch is obtained in an automated manner from the Fourier transform of the FLIC intensity of the speckles. If the rotational motion is not periodic there are no clear peaks observed in the frequency space and we ignore the event. At 0.6 nM Ncd almost none of the microtubules rotated clearly while at 6 nM Ncd many good events were observed (which is the reason why N is so small at 0.6 nM). In reality, a lot more experiments were performed at 0.6 nM in search for a few good events.

In Supplementary Figure 4C we explicitly showed one of the initially not rotating cases. However, showing the intensity plot corresponding to one of the speckles at 0.6 nM Ncd was misleading, as we did indeed see intensity fluctuations but they were not periodic and the intensity of different speckles did not correlate (which would not be the case if there was clear periodic rotation of the gliding microtubule).

Action taken: To make the point clearer, we have now supplemented the intensity plots with power spectral analysis.

Response: Moreover, our observations may be an indicator that the magnitude of torque generated scales with motor number/density. At 0.6 nM Ncd the torque is low and cannot overcome the friction generated by surface interactions while at 6 nM Ncd the torque is higher, and in most cases, friction is easily overcome.

Action taken: We have added new text to the manuscript to address the above-mentioned points.

On the same note both Figure 4 and Supplementary Figure 4 lack p-values.

Response: Thank you for pointing this out.

Action taken: P-values, obtained from Mann-Whitney U test, are now stated in a table included in Supplementary Figure 4.

The mention of chirality observed in yeast spindles in my opinion also merits a mention in the discussion, specifically Winey et al. J. Cell Biol 1995; Ding et al. J Cell Biol. 1993.

Response: Thanks a lot for this suggestion. We were indeed not aware of these papers.

Action taken: We now mention this work in the introduction/discussion.

Supplementary Movie 3 is not referenced in the main text.

Response: Again, thank you for pointing it out.

Action taken: We now refer to it in the paper.

Supplementary Figure 4E is referenced in the text, yet there is no Supplementary Figure 4E.

Response: This was a typo. Should be Supplementary Figure 4C.

Action taken: Corrected.

REVIEWERS' COMMENTS:

Reviewer #1 (Remarks to the Author):

The revised manuscript by Mitra et al. has now been greatly improved by editing and new data. The authors now demonstrated that Ncd can actually generate torque that is large enough to cause microtubule twisting and coiling. Although the data is qualitative, this demonstration is highly relevant to the chirality of the mitotic spindle in vivo. All comments of me have been addressed in great detail to the full satisfaction.

Reviewer #2 (Remarks to the Author):

I have read the answers of the authors as well as new data and find it to be satisfactory. I would also like to thank the authors for invested time and effort to improve on their work. I have no further objection for the article to be published in Nature Communications.

Response to Reviewers Comments -

REVIEWERS' COMMENTS:

Reviewer #1 (Remarks to the Author): The revised manuscript by Mitra et al. has now been greatly improved by editing and new data. The authors now demonstrated that Ncd can actually generate torque that is large enough to cause microtubule twisting and coiling. Although the data is qualitative, this demonstration is highly relevant to the chirality of the mitotic spindle in vivo. All comments of me have been addressed in great detail to the full satisfaction.

Reviewer #2 (Remarks to the Author): I have read the answers of the authors as well as new data and find it to be satisfactory. I would also like to thank the authors for invested time and effort to improve on their work. I have no further objection for the article to be published in Nature Communications.

Response: We thanks the reviewers for their positive evaluation of our work.